# Effects of Language Ontology on Transatlantic Automatic Speech Understanding Research Collaboration in the Air Traffic Management Domain

Shuo Chen [1,*], Hartmut Helmke [2], Robert M. Tarakan [1], Oliver Ohneiser [2], Hunter Kopald [1] and Matthias Kleinert [2]

1. The MITRE Corporation, 7515 Colshire Dr, McLean, VA 22102, USA; rtarakan@mitre.org (R.M.T.); hkopald@mitre.org (H.K.)
2. German Aerospace Center (DLR) Braunschweig, Institute of Flight Guidance, Lilienthalplatz 7, 38108 Braunschweig, Germany; hartmut.helmke@dlr.de (H.H.); oliver.ohneiser@dlr.de (O.O.); matthias.kleinert@dlr.de (M.K.)
* Correspondence: chen@mitre.org

**Abstract:** As researchers around the globe develop applications for the use of Automatic Speech Recognition and Understanding (ASRU) in the Air Traffic Management (ATM) domain, Air Traffic Control (ATC) language ontologies will play a critical role in enabling research collaboration. The MITRE Corporation (MITRE) and the German Aerospace Center (DLR), having independently developed ATC language ontologies for specific applications, recently compared these ontologies to identify opportunities for improvement and harmonization. This paper extends the topic in two ways. First, this paper describes the specific ways in which ontologies facilitate the sharing of and collaboration on data, models, algorithms, metrics, and applications in the ATM domain. Second, this paper provides comparative analysis of word frequencies in ATC speech in the United States and Europe to illustrate that, whereas methods and tools for evaluating ASRU applications can be shared across researchers, the specific models would not work well between regions due to differences in the underlying corpus data.

**Keywords:** automatic speech recognition; natural language understanding; semantic interpretation; air traffic control; radio communications; intent representation; semantic ontology; performance metrics

## 1. Introduction

### 1.1. Broad Context of the Study

For more than a decade, researchers in the United States and Europe have been developing and proving the benefit of Automatic Speech Recognition (ASR) applications in the Air Traffic Management (ATM) domain. In the United States, in support of the Federal Aviation Administration (FAA), the MITRE Corporation (MITRE) has developed capabilities to use Air Traffic Controller (ATCo)–pilot voice communication information for operational purposes, such as notifying ATCos of unsafe situations or analyzing operations to identify opportunities for safety or efficiency improvements. In Europe, as part of the Single European Sky ATM Research (SESAR) program, the German Aerospace Center (DLR) has led the development and testing of prototypic applications to enhance ATCo automation interactions, reduce ATCo workload, and identify safety issues in real time. Both MITRE [1] and DLR [2] have investigated the potential for automatic detection of readback errors, which are pilot errors in reading back ATCo instructions.

Key to most applications of ASR is the semantic meaning of the words spoken and transcribed, specifically in the context of the application in which the information will be used. Thus, we use the term Automatic Speech Recognition and Understanding (ASRU) to describe the speech-to-text and the text-to-meaning processes as one. ASRU for the Air

Traffic Control (ATC) domain needs to transcribe domain-specific words and phrases and then interpret their ATC meaning. For example, "lufthansa three twenty one one seventy knots until four contact tower eighteen four five" needs to be understood to capture the flight's callsign (DLH321) and the instructions it received (speed 170 knots until four miles from the runway; contact the tower on this radio frequency 118.450).

To represent the information contained in the speech—both the words and their semantic meaning in the ATC context—MITRE and European stakeholders, led by DLR, independently developed ATC language ontologies in support of ATM application development. A common ontology, used in both Europe and the US, could enable better sharing and reuse of data, models, algorithms, and software between the US and Europe.

In a recent paper [3], we described our collaboration to compare ontologies and identify opportunities for improvement and harmonization. This paper expands on that topic to discuss the impact of the ontology on future research and development collaboration, describing several ways that an ATC ontology is critical to facilitating collaboration between researchers and to appropriately evaluating ASRU applications in the ATM domain. This paper also examines the word-level differences between United States and European ATC speech to provide quantitative understanding of the corpus data that feed the ASRU models, informing their potential cross-use between regions. The analysis shows that whereas the methods and tools for developing and measuring ASRU performance can be shared across regions (e.g., between the US and Europe), the specific models built for the different regions would likely not work well across regions.

### 1.2. Structure of the Paper

This paper expands on the ontology study described in [3]. The following sections are organized as follows. Section 1.3 summarizes the uses of ASRU in ATC to date. Section 1.4 lays out the levels of an ontology in the context of the ATC domain. Section 2 presents two different concrete instantiations of ATC ontology and recalls examples presented in [3] that illustrate representations of ATC semantics using these ontologies. Section 3 describes the value of ATC ontology in facilitating collaboration between research groups and presents specific applications and the semantic representations they rely on. Section 4 presents a quantitative comparison of ATC speech at the word level between the United States and Europe. Finally, Section 5 completes the paper with our conclusions and next steps.

### 1.3. Background

Voice communications are an essential part of ATC because they are the primary means of communicating intention, situation awareness, and environmental context. Over the last decade, researchers have invested tremendous effort into advancing the accuracy and sophistication of in-domain ASR and Natural Language Understanding (NLU) capabilities to enable human–machine teaming that improves aviation safety and efficiency [4].

Early applications of ASR and NLU focused on simulation pilots for high-fidelity controller training simulators because these applications were in controlled environments with well-defined phraseology and a limited set of speakers [5–7]. Other examples for replacing pseudo-pilots in training environments are from the FAA [8,9], DLR [10], and DFS [11]. Later applications in lab settings expanded to simulation pilots for human-in-the-loop simulations in ATM research measuring workload [12]. With the adoption of electronic flight strips in ATC facilities, Helmke et al. [13] applied ASRU to demonstrate the effectiveness of speech assistants in reducing controller workload and improving efficiency. Prototypes demonstrating the use of ASRU to enhance safety in live operations also emerged. ASRU can support the detection of anomalous trajectories [14]. It can also support the detection of closed runway operations and wrong surface operations in the tower domain [15]. The efficacy of using ASRU to automatically detect readback discrepancies was analyzed in the US [1] and in Europe [2]. A safety monitoring framework that applied ASR and deep learning to flight conformance monitoring and conflict detection has been proposed by [16]. The growing prevalence of uncrewed aerial vehicles has also led

to use cases in autonomous piloting. Text-to-speech and NLP can enable communications between human controllers and autonomous artificial intelligence pilots as advocated by [17]. Finally, the accuracy and robustness achieved by mature in-domain ASR has enabled mining of large-scale ATC communication recordings for post-operational analyses. Chen et al. [18] measured approach procedure usage across the U.S. National Airspace System using automatically transcribed radio communications in post analyses. Similarly, reference [19] assessed the quantity of pilot weather reports delivered over the radio against the quantity of pilot reports manually filed during the same time frame.

A common theme across all these applications is the use of a language understanding layer that distills and disambiguates semantic meaning from the text transcripts generated by ASR. Although there is variability in the semantic structures and concepts relevant to each use case, almost all extracted semantics relate to the representation of controller and pilot intent or situation awareness. Currently, research groups in the US and Europe create and maintain their own semantic taxonomies or ontologies to define the elements and relationships that represent intentions or situational context relevant to their specific use cases. These elements usually cover ATC concepts such as aircraft callsigns, command types, and command values in a structured human-readable and machine-readable formats.

The European ontology was defined by fourteen European partners from the ATM industry as well as by air navigation service providers (ANSPs) funded by SESAR 2020 [20]. The ontology was refined through use by different projects, such as STARFiSH [21], "HMI Interaction Modes for Airport Tower" [22,23] in the tower environment, "HMI Interaction modes for approach control" [24], and HAAWAII [25], which expanded the ontology to support pilot utterances [2].

The MITRE ontology was developed and matured over several years, with many contributing projects. Our earliest ontology was created for the simulation pilot component of an enroute ATCo trainer [5]. It was later expanded to incorporate tower domain phraseology for projects such as the Closed Runway Operations Prevention Device [15]. More recently, to support the varied use cases required of our large-scale, post-processing capability [18], the ontology was expanded to cover most of the phraseology for the standard operations documented in [26]. With each iteration we made it more robust and flexible to cover regional phraseology variations across the operational domains, i.e., tower, terminal, and enroute airspace.

### 1.4. What We Mean by Ontology

An ontology in the context of this paper is a collection of rules, entities, attributes, and relationships that define how language meaning is represented in a particular domain. An ontology introduces structure to ASRU by distinguishing between the four levels of language communication—lexical, syntactical, semantic, and conceptual [27]—and defining meaning representation within these levels.

The **lexical level** deals with words and distinguishes between synonyms—words with the same meaning that are spoken differently. For example, the words *nine* and *niner* denote the same numerical value in the ATC domain. Similarly, *speed bird* and *speedbird* signify the same commercial airline. The ontology rules at this level specify the universe of words (i.e., the vocabulary) that may appear in ATC radio communications.

The **syntactical level** deals with grammar and distinguishes between similar meaning phrases that are worded differently. For example, the phrases *runway two seven left cleared to land*, and *cleared to land two seven left* are syntactically different because they have different word ordering; however, they have the same meaning, which is to convey clearance to land on runway two seven left.

The **semantic representation level** deals with meaning despite differences in vocabulary or grammar that do not affect the meaning of the communication. The ontology rules at this level may deal with meaning that is explicitly spoken as well as meaning that is implied. Both phrases from the syntactical level example may be mapped to an agreed form

such as CTL RWY 27L or RW27L CLEARED_TO_LAND. Later in this paper, we discuss how these semantics are represented in the European and MITRE ontologies.

The **conceptual level** deals with a higher level of understanding that goes beyond the semantic level. It captures the bigger picture, which in the ATC domain can be bigger than the sum of the individual radio transmissions. An example of an event at the conceptual level is the concept of an aircraft being in the arrival phase of flight. For some applications, this is more important than knowing the particular set of altitude and speed reductions an ATCo issued. Another example is the speech associated with a go-around, which might involve a back-and-forth discussion between an ATCo and pilot followed by a series of ATCo instructions.

In this paper, the ontology instantiations we describe primarily address the lexical and semantic level described above. However, we believe ontologies can and should expand to cover any information that is relevant to the application using language interpretation.

## 2. A Comparison of Two ATC Ontologies

This section recaps the comparison of US and European ATC ontology instantiations described in [3].

### 2.1. Lexical Level

At the lexical level, MITRE's ontology specifies that both speech and non-speech sounds during ATC radio communications should be captured in the transcription. Furthermore, the transcription should closely represent the sounds present in the audio without additional annotation or meaning inference. This means speaker hesitation sounds such as "um" and "uh", partially spoken words, foreign words such as "bonjour" or "ciao", and initialisms such as "n d b" and "i l s" are transcribed as they sound. These rules were based on best practices in automatic speech recognition training corpus creation.

The European ontology at the lexical level requires that both speech and non-speech sounds be annotated in the transcription. Special annotation is associated with non-English words spoken in a radio transmission to indicate non-English content. Domain-specific acronyms and initialisms such as "NDB" and ILS" are transcribed as words in the vocabulary. Special handling is associated with domain-specific synonyms such as "nine" and "niner", which are transcribed to a single lexical representation, "nine". Both ontologies stick to the standard 26 letters in the English alphabet, i.e., "a" to "z" in lower- and upper-case form. Diacritical marks such as the umlaut "ä" in German or the acute accent "é" in French are not supported.

The differences that we observed at the word level can be summed up as fitting into the following categories:

- Identical words with different spellings (e.g., *juliett* versus *juliet*).
- How initialisms are handled (e.g., *ILS* versus *i l s*).
- Words with similar meaning and different pronunciations and spelling (e.g., *nine* versus *niner*).
- Words absent from one ontology or the other (e.g., the word *altimeter* does not occur in European ATC communications and the corresponding ICAO term *QNH* is absent from US ATC communications) [28].
- Whether speech disfluencies and coarticulation are captured at the word level (e.g., *cleartalan* versus *cleared to land*).
- Words not represented in the US English language (e.g., the German word *wiederhoeren* for a farewell).

These differences can have an impact on ASR speed and accuracy performance and on the end user or downstream software application.

### 2.2. Semantic Level

At the semantic level, MITRE's ontology ($SL_{US}$) specifies a set of entities, attributes, and relationships that capture meaning at the command or clearance level. Figure 1 illustrates

the ontology of SL$_{US}$ in graph format. At the highest level, SL$_{US}$ starts with a concept called *Command Interpretation* that represents an instruction, and it has a mandatory attribute called *Command Type*. The *Command Type* attribute declares the type of the instruction, such as an aircraft maneuver such as "climb" or a clearance to fly a procedure such as "cleared ILS two one approach".

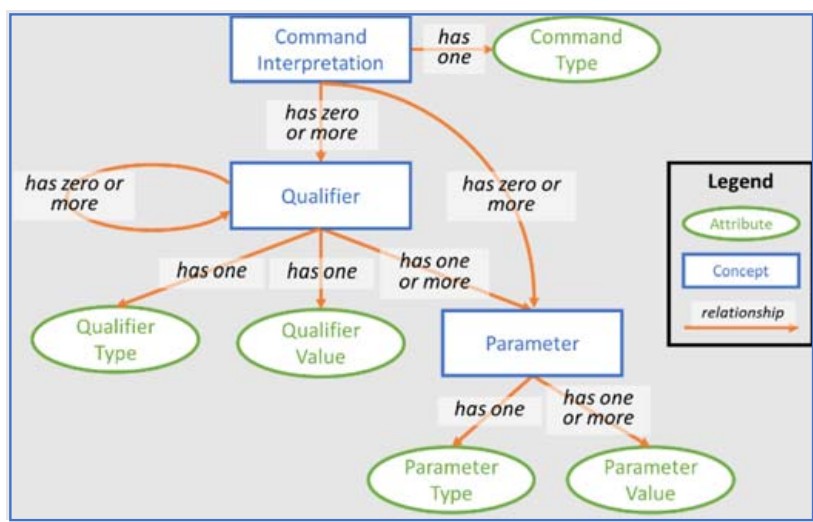

**Figure 1.** Graphical representation of SL$_{US}$ ontology.

Each *Command Interpretation* can have zero or more child concepts called *Qualifiers* and *Parameters*. Both characterize, modify, and/or add values to the instruction. *Qualifiers* disambiguate or characterize *Parameters* by representing value units that are lexically present in the transcript, e.g., "flight level", "heading", "knots", etc. *Qualifiers* can be nested to represent deeper, hierarchical relationships. For example, to represent the condition "until the dulles VOR", the highest-level *Qualifier* would represent the preposition "until", its child *Qualifier* would represent the waypoint type "VOR", and its child *Parameter* would represent the name of the waypoint "dulles".

*Parameters* represent the value payloads for instructions that require a value, such as a heading (in degree) for a turn instruction or an altitude (in feet or flight level) for a climb instruction. A *Parameter* may exist without a *Qualifier* parent if the format of the *Parameter* value or the instruction's command type makes the *Parameter* inherently unambiguous. For example, in the instruction "climb three four zero", the command type "climb" allows us to infer that an altitude must be represented in the *Parameter* and the value format in three digits allows us to infer that the altitude is in flight level even though a unit is not explicitly stated. Figure 2 illustrates the SL$_{US}$ ontology as a block diagram for comparison with the semantic level of the European ontology in Figure 3.

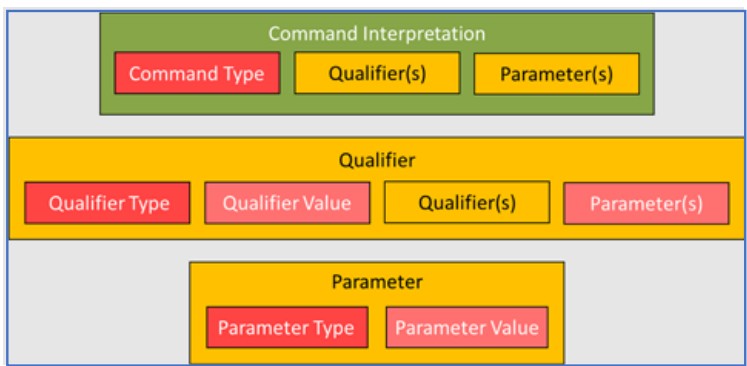

**Figure 2.** Block diagram of SL$_{US}$ ontology; optional elements in orange.

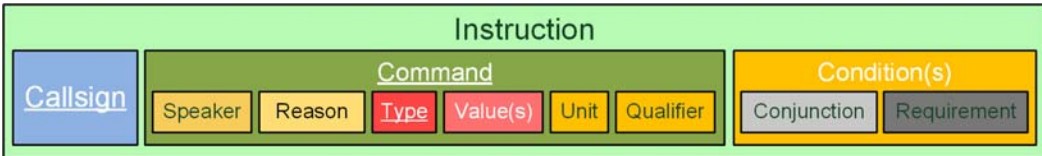

**Figure 3.** Block diagram of SL$_{EU}$ ontology; optional elements in orange.

In comparison, Figure 3 illustrates the semantic level of the European ontology (SL$_{EU}$). At its highest level, SL$_{EU}$ starts with a concept called *Instruction*, i.e., a mandatory *Callsign*, a mandatory *Command*, and optional *Conditions*. If the *Callsign* cannot be extracted from the transmission, the *Callsign* is "NO_CALLSIGN". A *Command* concept always has a *Type* attribute that declares the type of instruction represented. When no *Command* is found in a transcript, a *Command* concept with *Type* "NO_CONCEPT" is created. Depending on the *Type*, no *Value* or one or more *Values* can follow. If a *Value* is available, the optional attributes *Unit* and *Qualifier* are possible. The optional *Condition* concept can be present for any *Type* and more than one may be associated with one *Command*.

*Type* can consist of a subtype, as illustrated by the command CLEARED ILS. The Speaker attribute can have the values "ATCO" or "PILOT". If not specified, it is ATCO or can be derived from additional available context information. The *Reason* attribute is only relevant for pilot transmissions. Then the values "REQ=REQUEST", "REP=REPORTING", or an empty value are possible. The empty value, i.e., the default value, in most cases contains a pilot's readback. The *Reason* attribute is motivated by the examples in Table 1.

**Table 1.** Examples of ontology representations on ATC communications transcripts.

| | | |
|---|---|---|
| Lexical Representation | eurowings 1 3 9 alpha cleared I L S approach oh 8 right auf wiedersehen | |
| | MITRE Ontology Word-Level | eurowings uh one three niner alfa cleared i l s approach oh eight right auf wiedersehen |
| | European Ontology Word-Level | euro wings [hes] one three nine alfa cleared ILS approach [spk] O eight right [NE German] auf wiedersehen [/NE] |
| Controller Transmission | november three mike victor cleared I L S runway two one approach | |
| | SL$_{US}$ | Callsign: {N, 3MV, GA}, Cleared: {21, ILS} |
| | SL$_{EU}$ | N123MV (CLEARED ILS) 21 |
| Callsign and Unit Inference | fedex five eighty two heavy maintain four thousand three hundred | |
| | SL$_{US}$ | Callsign: {FDX, 582, H, Commercial}, Maintain: {Feet, 4300} |
| | SL$_{EU}$ | FDX482 (MAINTAIN ALTITUDE) 4300 none |
| Transmission with Multiple Commands | good day american seven twenty six descend three thousand feet turn right heading three four zero | |
| | SL$_{US}$ | Callsign: {AAL, 726, Commercial}, Courtesy, Descend: {3000, Feet}, TurnRight: {340, Heading} |
| | SL$_{EU}$ | AAL726 GREETING, AAL726 DESCEND 3000 ft, AAL726 HEADING 340 RIGHT |
| Transmissions without Callsign | fly zero four zero cleared I L S approach | |
| | SL$_{US}$ | Fly: {040, Heading}, Cleared: {ILS} |
| | SL$_{EU}$ | NO_CALLSIGN HEADING 040 none, NO_CALLSIGN (CLEARED ILS) none |
| | lufthansa one two charlie go ahead | |
| | SL$_{US}$ | Callsign: {DLH, 12C, Commercial} |
| | SL$_{EU}$ | DLH12C NO_CONCEPT |

**Table 1.** *Cont.*

| | | |
|---|---|---|
| | lufthansa six alfa charlie descend one eight zero break break speed bird six nine one turn right heading zero nine five cleared I L S runway three four right | |
| Transmissions with more than one Callsign | SL$_{US}$ | Callsign: {DLH, 6AC, Commercial}, Descend: {180, FL}, Callsign: {BAW, 691, Commercial}, TurnRight: {95, Heading} Cleared: {34R, ILS} |
| | SL$_{EU}$ | DLH6AC DESCEND 180 none, BAW691 HEADING 095 RIGHT, BAW691 (CLEARED ILS) 34R |
| | stand by first speed bird sixty nine thirteen turn right by ten degrees | |
| | SL$_{US}$ | Callsign: {BAW, 6913, Commercial}, TurnRight: {10, Degrees} |
| | SL$_{EU}$ | NO_CALLSIGN CALL_YOU_BACK, BAW6913 TURN_BY 10 RIGHT |
| Altitude with Limiting Condition | maintain four thousand feet until established | |
| | SL$_{US}$ | Maintain: {4000, Feet} |
| | SL$_{EU}$ | (MAINTAIN ALTITUDE) 4000 ft (UNTIL ESTABLISHED) |
| Instructions with Position-Based Conditions | at dart two you are cleared I L S runway two one left | |
| | SL$_{US}$ | Cleared: {21L, ILS} |
| | SL$_{EU}$ | NO_CALLSIGN (CLEARED ILS) 21L (WHEN PASSING DART2) |
| | leaving baggins descend and maintain one four thousand feet | |
| | SL$_{US}$ | Descend: {14,000, Feet} |
| | SL$_{EU}$ | NO_CALLSIGN DESCEND 14,000 ft (WHEN PASSING BGGNS) |
| Instructions with Advisories | maintain two fifty knots for traffic | |
| | SL$_{US}$ | Maintain: {250, Knots, for traffic} |
| | SL$_{EU}$ | NO_CALLSIGN (MAINTAIN SPEED) 250 kt, NO_CALLSIGN (INFORMATION TRAFFIC) none |
| | traffic twelve o'clock two miles same direction and let's see the helicopter | |
| | SL$_{US}$ | Traffic: {Distance: 2, OClock: 12, TrafficType: helicopter} |
| | SL$_{EU}$ | NO_CALLSIGN (INFORMATION TRAFFIC) |
| | caution wake turbulence one zero miles in trail of a heavy boeing seven eighty seven we'll be going into this [unk] | |
| | SL$_{US}$ | Wake: () |
| | SL$_{EU}$ | (CAUTION WAKE_TURBULENCE) |
| Pilot Transmission as Readback | descend flight level one seven zero silver speed | |
| | SL$_{US}$ | Descend: {170, FL} |
| | SL$_{EU}$ | NO_CALLSIGN PILOT DESCEND 170 FL |
| Pilot Transmission as Report | speed bird two one alfa flight level two one two descend flight level one seven zero inbound dexon | |
| | SL$_{US}$ | Callsign: {BAW, 21A, Commercial} Descend: {170, FL} |
| | SL$_{EU}$ | BAW21A PILOT REP ALTITUDE 212 FL BAW21A PILOT REP DESCEND 170 FL BAW21A PILOT REP DIRECT_TO DEXON none |
| Correction of Instruction | speed bird one one descend level six correction altitude six thousand feet | |
| | SL$_{US}$ | Callsign: {BAW, 11, Commercial}, Descend: {6000, Feet}, |
| | SL$_{EU}$ | BAW11 CORRECTION BAW11 DESCEND 6000 ft |
| | speed bird one one descend level six correction six thousand feet disregard turn left heading three two five degrees | |
| | SL$_{US}$ | Callsign: {BAW, 11, Commercial} Descend: {6000, Feet} TurnLeft: {325, Heading} |
| | SL$_{EU}$ | BAW11 DISREGARD BAW11 HEADING 325 LEFT |

The differences that we observed between $SL_{US}$ and $SL_{EU}$ at the semantic level can be summed up as fitting into the following categories:

- How callsigns are represented.
- The extent of and representation of inferred and implied information in the semantic representations.
- The level of detail represented for advisory-type transmissions (e.g., traffic advisories, pilot call-in status information).
- Which less-common ATCo instructions have defined representations.
- How ambiguous ATCo instructions are represented.

For a detailed comparison of the semantic-level ontology overlap between the MITRE and European ontology instantiations, refer to Tables A1–A6 in the Appendix A.

### 2.3. Examples of Ontology Representations from ATC Communications

In reference [3], we presented several examples of word-level and semantic interpretation representations as defined by the European and MITRE ontology instantiations. We summarize them again below in Table 1 to illustrate the similarities and differences between the two ontology instantiations.

### 2.4. Quantifying the Differences

MITRE and DLR each exchanged 100 transmissions, with transcripts and semantic annotations, from the terminal area of a major US airport and a European hub airport. The US transcripts and annotations were manually transformed into the European format and vice versa. We assessed the word-level differences at the transcript level in terms of Levenshtein distance [29].

Out of 1554 total words in transmissions, 187 of them required modification to adhere to the other party's ontology, i.e., 12.0% of words were modified through substitution (89), deletion (35), and insertion (63). We omit uppercase to lowercase transformation from this measure. Figure 4 shows a sample transcript and its transformation.

```
all · · · · right cleared for the ils · · · · two five ·
alright · · · · · cleared for the i · 1 s two five ·
```

**Figure 4.** $SL_{EU}$ structure for ambiguous instructions. Word-level difference between European (first row) and US (second row) transcripts resulting in a Levenshtein distance of 5.

In the following bullets, we list and explain some of the most often occurring cases from the 200 transcripts that are represented differently at the word level in the MITRE and European ontologies as sketched in Section 2.3:

- Separation and combination of words/letters
  - "ILS" vs. "i l s" (23 times)
  - "southwest", etc., vs. "south west", etc. (19 times)
- Different spellings
  - "nine" vs. "niner" (9 times)
  - "juliett" vs. "juliet" (6 times)
  - "OK" vs. "okay" (4 times)
- Special sounds and their notation
  - "[unk]" vs. no transcription (7 times)
  - "[hes]" vs. "uh" (7 times)

Table 2, taken from [3], shows the overlap in commands represented by the MITRE and European ontologies at the semantic level after analyzing 121 ATCo instructions from Europe and 120 from the US. DESCEND in $SL_{EU}$ corresponds to Descend in $SL_{US}$. MAINTAIN ALTITUDE with Value and Unit in $SL_{EU}$ corresponds to Maintain with the US Qualifier feet or FL. The Cleared ILS Z in $SL_{US}$ now corresponds to CLEARED ILSZ in

SL$_{EU}$. GREETING and FAREWELL in SL$_{EU}$ correspond to Courtesy in SL$_{US}$. SL$_{US}$'s Radar Service Terminated is currently not modelled in SL$_{EU}$. In contrast, SL$_{US}$ does not model SL$_{EU}$'s CALL_ YOU_BACK command type.

**Table 2.** Percentage of overlap based on analysis of 241 ATCo instructions.

| Type of Semantic Comparison | Overlap of Concepts |
|---|---|
| Concept present in both ontologies before adaptation | 82% |
| Corresponding concept after small adjustments | 95% |
| Achievable match with existing model structures | 100% |

## 3. Impact of Ontology on Collaboration

Up to this point in the paper, we have described and compared two ontology instantiations that define simplifying meaning representations for ATC communications. In the remainder of this paper, we will describe how these ontologies assist collaboration, highlighting their benefits and shortfalls. Specifically, we examine the extent to which data, models, algorithms, and applications can be shared between research groups given operational and geographic differences and how the differences manifested in ATC communications can be bridged with the help of ontologies.

### 3.1. Data Sharing

#### 3.1.1. Text Data

In a perfect world, there would exist only one ground truth transcript for a segment of speech audio. However, as the ontology differences summarized above show, even when there is agreement on what was spoken, lexical representation of the spoken content can still differ. Although these differences in representation may seem superficial, they leave lasting impressions on models created using these lexical representations and can lead to artificially inflated error metrics if overlooked and in some cases can increase the number of actual errors.

For example, consider the two nominal examples in Table 3, where the original ground truth transcripts are transcribed according to the European ontology rules and the automatically transcribed text is generated by a speech recognizer that has modeled language following the MITRE ontology rules.

**Table 3.** Nominal examples of transcription error without lexical translation.

| | Ground Truth Transcript | Automatically Transcribed Text |
|---|---|---|
| Example 1 | good day american seven twenty six descend three thousand feet turn right heading three four zero | good day american seven twenty six descend four thousand feet turn right heading three zero |
| Example 2 | cleared ILS three four | cleared i l s three five |

In Example 1, when the automatically transcribed text is assessed against the ground truth transcript using word error rate (WER), a common metric for assessing speech recognition accuracy, the WER evaluates to 12.5% because of one substitution error (three by four) and one deletion error (four is missing) against a total of 16 words in the ground truth. This WER is reasonable because in this scenario the ground truth and the speech recognizer have the same lexical representation for all words in the transcript.

In contrast, in Example 2, the three errors (1 substitution and 2 insertions) resulting from differences in lexical representation ("i l s" instead of "ILS") compound the actual substitution error ("four" by "five") and results in a WER of 100%. In this scenario, lexical differences artificially inflate the true WER from 25% to 100%. Furthermore, if the semantic parser does use the same lexical representation, the difference can lead to parse errors, which in turn lead to semantic errors.

Thus, a mechanism for translation between different lexical representations is often required when sharing raw text data. By explicitly defining the rules for lexical represen-

tation, ontologies play a critical role in highlighting what is required of the translation process and facilitate its design without extensive data analysis and exploration. Because WER is an indicator of lexical representation mismatch, it can be repurposed to measure the effectiveness of the translation process.

### 3.1.2. Semantic Annotations

Semantic representation differences are often much more obvious than lexical representation differences but they still require the same, if not more, attention to translation. The complexities of semantic representation make ontologies even more critical to the translation design process. Though an exhaustive comparison of ontology instantiations may seem daunting, it is still much easier than an exhaustive search for syntactic and semantic samples in raw text data!

As in the case with lexical translation, a measure of semantic representation mismatch is needed to assess effectiveness of the translation process. We outline below our simple scheme for measuring semantic translation accuracy that is independent of semantic concept type or subcomponents and treats all semantic components with equal importance. These metrics can be used to compare semantic labels that have been mapped from one ontology representation to another and then back again to assess semantic content loss from the conversion. Table 4 lists definitions that are the building blocks for the accuracy metrics, and Table 5 defines the metrics and their formulas.

**Table 4.** Definition of basic element for accuracy calculation.

| Name | Definition |
|---|---|
| True Positive (TP) | TP is the total number of True Positives: The concept is present and correctly and fully (including all subcomponents) detected |
| False Positive (FP) | FP is the total number of False Positives: The concept is incorrectly detected, i.e., either the concept is not present at all or one or more of its subcomponents are incorrect |
| True Negative (TN) | TN is the total number of True Negatives: The concept is correctly not detected |
| False Negative (FN) | FN is the total number of False Negatives: A concept is not detected when it should have been |
| Total (TA) | TA is the total number of annotated transcripts, i.e., the number of gold transcripts |

**Table 5.** Accuracy metrics for semantic representations.

| Name | Definition |
|---|---|
| Recall | $\frac{TP}{TP+FN}$ |
| Precision | $\frac{TP}{TP+FP}$ |
| Accuracy | $\frac{TP+TN}{TP+TN+FP+FN}$ |
| $F_1$-Score | $\frac{2 * Recall*Precision}{Recall+Precision}$ |
| $F_\alpha$-Score | $\frac{(1+\alpha^2) * Recall*Precision}{(\alpha^2*Precision)+Recall}$ |
| Command Recognition Rate (RcR) | $\frac{TP+TN}{TA}$ |
| Command Recognition Error Rate (CRER) | $\frac{FP}{TA}$ |
| Command Rejection Rate (RjR) | $\frac{FN}{TA}$ |

Consider the nominal example of semantic translation for the transcript in Table 6, "*good day american seven twenty six descend three thousand feet turn right heading three four zero*". We use this example to illustrate the metrics in action.

**Table 6.** Nominal example of semantic translation.

| Ground Truth Semantics | Translated Semantics |
|---|---|
| AAL726 GREETING, | AAL726 GREETING, |
| AAL726 DESCEND 3000 ft, | AAL726 DESCEND, |
| AAL726 HEADING 340 RIGHT | AAL726 HEADING 340 RIGHT |

In this example, there are 2 TPs (greeting and heading change) and 1 FP (due to the missing altitude in the altitude change), 0 FN, 0 TN, and TA = 3. Table 7 summarizes the accuracy metrics calculated on this nominal example.

**Table 7.** Accuracy metrics calculated on nominal example of semantic translation.

| Name | Definition | Example |
|---|---|---|
| Recall | 2/(2 + 0) | 100% |
| Precision | 2/(2 + 1) | 66% |
| Accuracy | (2 + 0)/(2 + 0 + 1 + 0) | 66% |
| F1-Score | $F_1 = \frac{2*100\%*33\%}{100\%+33\%}$ | 50% |
| Command Recognition Rate (RcR) | (2 + 0)/3 | 66% |
| Command Recognition Error Rate (CRER) | 1/3 | 33% |
| Command Rejection Rate (RjR) | 0/3 | 0% |

The range for all metrics, with the exception of the Command Recognition Error Rate (CRER), is between 0 and 1. The CRER could go above 1 if (many) concepts not present in the ground truth are generated.

These metrics provide a general measure of the semantic coverage overlap between ontologies, i.e., when there is significant overlap, the CRER is low and when there is little overlap, the CRER is high. These same metrics can measure the extraction accuracy of a rules-based or deep neural network semantic parser in a general sense, but they should be modified and supplemented before use as a measure of application accuracy performance. We detail the rationale for and examples of application-specific metrics later in this section.

### 3.2. Reusing Models and Algorithms

#### 3.2.1. Automatic Speech Recognition Models

In today's world of large pre-trained models, automatic speech recognition models are usually robust enough to transplant into new geographic regions, environments, and domains with minimal finetuning. Some models can even adapt to language changes with little to no finetuning! However, there are idiosyncrasies in the ATC language that can reduce a speech recognition model's performance if they are not addressed during transplantation between geographic regions or simply throughout prolonged use. Specifically, the quantity of airspace and region-specific, i.e., site-specific, proper nouns used during ATC radio communications requires special handling and maintenance when operating a speech recognition model in the ATC domain.

A lot of the vocabulary that appears in ATCo–pilot communications includes general purpose words such as climb, descend, cleared, to, for, and, until, one, two, three, alfa, bravo, and charlie. These are simple to document in the word level of an ontology. However, depending on the quantity of airspace that the ASRU is intended to cover, a significant percentage (90% or more) of the vocabulary could be made up of names, such as those for airline callsigns, facility identifiers, location identifiers, navigational aids, and procedure identifiers.

The site-independent, general-purpose vocabulary is relatively static and short—just a few hundred words covers most ATCo–pilot voice communications. Section 4 will show that 551 words cover 95% of the spoken words in the US data. The vocabulary of names that are used in ATCo–pilot voice communications is much larger (tens of thousands if covering the entire United States airspace) and subject to change to accommodate airspace and procedure revisions and airline and pilot callsign name additions. This name list is

disproportionately large compared with the general-purpose word list but not excessively large by ASR standards. More importantly, the list of names is much more dynamic, which creates a challenge. Just as software can deteriorate over time (i.e., software rot), ASRU ontologies (and their associated models) can degrade over time if they are not maintained. For ASRU applications, an outdated word-level ontology is likely to result in out-of-vocabulary errors, which can negatively affect ASR accuracy and the accuracy of all downstream capabilities. The same applies for the sequence of words, i.e., the ICAO-phraseology and the deviation from ICAO phraseology [28]. This is a serious lifecycle maintenance issue. It is a particularly large challenge for applications that need to be scaled-up to cover multiple ATC sectors and facilities. Newer ASR models, which transcribe at the letter level, and language model tokenizers, which tokenize at the subword level, may eliminate the problem of "out-of-vocabulary" words but not the challenge of correctly recognizing and interpreting these words given their low occurrence in the training data. Furthermore, the unconstrained vocabulary in these models presents its own problems to interpretation.

Changes on the ATC operations side are made on the 28-day AIRAC (Aeronautical Information Regulation And Control) cycle. The number of changes during any one AIRAC cycle is usually small and the changes are known well in advance. Changes on the commercial airline side do not follow an official cycle but tend to be relatively uncommon. There are two subcategories of names that can present unique problems for ASRU: Military callsigns and five-letter waypoint names.

Military callsigns are a challenge because they can be introduced ad hoc and are not always known in advance of a flight's departure. The FAA ATC handbook [26] states that: *U.S. Air Force, Air National Guard, Military District of Washington priority aircraft, and USAF civil disturbance aircraft. Pronounceable words of 3 to 6 letters followed by a 1- to 5-digit number.* These pronounceable words may be pilots' names or nicknames and these words might not otherwise appear in an ATC ontology and associated ASR model. For example, "*honda five*" and "*maverick zero zero seven*" are examples of accepted military callsigns.

Five-letter waypoint names present a different challenge for ASRU. They are part of the AIRAC update cycle and are published in advance, but only the five-letter codes are published, not their pronunciations. In many cases, the waypoint codes correspond to obvious words or can be sounded out using a simple algorithm—but not always! For example: *GNDLF*, *YEBUY*, and *ISACE*. Whereas pronunciation can be handled manually on a small scale by talking to the ATC personnel for a facility for some applications, it does not easily scale to applications involving multiple ATC facilities or large amounts of airspace.

In ASRU, there is a fundamental tradeoff between a vocabulary that is too small, resulting in out-of-vocabulary errors, and a vocabulary that is too large, resulting in confusion between similar sounding words. An ASR built using a larger word-level ontology is not always better. Furthermore, it may not be possible to know and include the region-specific names in the vocabulary until you know the region where the model will be used. Thus, a word-level ontology may only specify the general-purpose vocabulary explicitly and define rules for how this vocabulary should be augmented with site-specific names before use. This issue contributes to the challenge of sharing ASR models trained and/or used between different ATC facilities or regions. Well-designed ASRU tools can simplify the adding of this site-specific information to the ontology and corresponding software.

### 3.2.2. Semantic Parsing Algorithms

Semantic parse algorithms translate lexical representations into semantic representations by capturing and translating the syntactical relationships between words. The mechanism for semantic parsing could be a rules-based algorithm or a machine-learning-based neural network model. Both are sensitive to lexical representation changes because they operate so closely on lexical and syntactic relationships.

Rules-based semantic parse algorithms could be considered a part of the ontology at the syntactical level because they contain rules about which relationships between lexical representations are meaningful and how they can be interpreted to construe higher-level semantic concepts. As every acceptable permutation of words must be explicitly or implicitly specified for interpretation, rules-based parse algorithms inherently document the syntactic level of the ontology; however, they can be incredibly labor intensive to create and maintain. Transplanting a rules-based semantic parse algorithm into a new region requires adapting the parse algorithm to regional lexicons, site-specific operational communications, and jargon. This inherently updates the syntactic level of the ontology as part of the model transition process.

Machine-learning-based models for semantic parsing learn the syntactic relationships from the hierarchies present in the semantic labels. In one sense, this eases the burden of rule creation, but it shifts it instead to data labeling, because the data labels must reflect the relationships between lexical entities in order for the model to learn them. Furthermore, as the syntactic rules are no longer explicitly stated as rules but hidden within the model weights, exact syntactic relationships can be difficult to discover and adjust for new model users, hampering reuse and even certification. In the absence of explicit syntactic rules, the semantic definitions of the ATC ontology become even more important as they capture and relay semantic hierarchies that might otherwise be overlooked without exhaustive data search and analysis.

### 3.3. Sharing and Reusing Applications

In ATC, there are common areas for improvement that come up again and again as possible avenues for ASRU application. As a result, the potential for application transition and reuse is high when an application is successful, even across geographic boundaries. In this section, we describe how ontologies facilitate application transition. We also discuss the importance of application-specific metrics and why they should be added to the ontology on an as-needed basis.

### 3.3.1. Examples of Application Specific Ontologies

Most applications incorporating ASRU are unlikely to use all the semantic concepts defined in an ATC ontology. Indeed, some of the applications prototyped between MITRE and DLR have only used a handful each. However, some semantic concepts appear across multiple applications, marking them as particularly important and worthy of focused research to improve extraction accuracy. Callsign is a recurring semantic concept that is relevant to multiple applications. Thus, both MITRE and DLR have special handling, such as context-based inference, to improve the detection accuracy of this concept.

Table 8 summarizes different applications of ASRU prototyped by DLR and MITRE. The table elucidates by an "X", which command semantics are used in each application. The applications are described in greater detail below the table and references of published reports are provided where available.

### 3.3.2. Closed Runway Operation Detection (CROD)

MITRE prototyped and field tested a closed runway operation clearance detection system that uses ASRU to detect landing or takeoff clearances to runways that are designated as closed. The system relies purely on manual entry of runway closures and passive listening on the local controller radio channel to detect a clearance to a closed runway and issue an alert. For more information on this application, please see [15].

### 3.3.3. Wrong Surface Operations Detection (WSOD)

An expansion on the closed runway operation clearance detection system, this more advanced prototype combines ASRU on radio communications with radar data in real time to detect discrepancies between the landing clearance runway issued over the radio and the

projected landing runway inferred from radar track data. When a discrepancy is detected, the system generates an alert to the tower ATCo.

**Table 8.** Semantic representations relevant to specific applications.

| Applications [1] / Command-Type Categories | CROD Closed Runway Operation Detection | WSOD Wrong Surface Operations Detection | ACUA Approach Clearance Usage Analysis | PRLA Prefilling Radar Labels—Approach | MRT Multiple Remote Tower Operations | SMGCS Integration with A-SMGCS—Apron | WLP Workload Prediction in London TMA | CPDLC Integration with CPDLC | PWR Pilot Weather Reports | VFR Use of Visual Separation | SPA Simulation Pilot—Apron | SPET Simulation Pilot—Enroute Training | RB-E Readback Error Detection—Enroute | RB-T Readback Error Detection—Tower |
|---|---|---|---|---|---|---|---|---|---|---|---|---|---|---|
| Acknowledgement | | | | | | | | | | | | X | X | |
| Airspace Usage Clearance | | | | | X | | | | | | | | X | |
| Altimeter/QNH Advisory | | | | X | | | | X | | | | X | X | |
| Altitude Change | | | | X | X | | X | X | | | | X | X | |
| Vertical Speed Instruction | | | | X | | | X | X | | | | X | X | |
| Attention All Aircraft | | | | | | | | | | | | X | | |
| Callsign | | X | X | X | X | X | X | X | X | X | X | X | X | X |
| Cancel Clearance | | | X | | | | | | | | | | | |
| Correction/Disregard | | | | X | X | X | X | X | | | X | | X | |
| Courtesy | | | | | | | X | | | | | | | |
| Future Clearance Advisory | | | | X | | | | | | | | | X | |
| Heading | | | | X | | | X | X | | | | X | X | |
| Holding | | | | X | X | | X | X | | | | X | X | |
| Information (Wind, Traffic) | | | | X | X | | | | | X | | | | |
| Maintain Visual Separation | | | | | | | | | | X | | | | |
| Pilot Report | | | | | | | | | X | | | | X | |
| Procedure Clearance | | | X | X | X | | | X | | | | X | | |
| Radar Advisory | | | | X | X | | | | | | | | | |
| Radio Transfer | | | | X | X | X | X | X | | | X | X | X | |
| Reporting Instruction | | | | X | X | X | X | | | | X | X | X | |
| Routing Clearance | | | | X | | | X | X | | | | X | X | |
| Runway Use Clearance | X | X | | X | X | X | | | | | X | | | X |
| Speed Clearance | | | | X | | | | X | | | | X | X | |
| Squawk | | | | | X | | | | | | | X | X | |
| Taxi/Ground Clearance | | | | X | X | | | | | | X | | | |
| Traffic Advisory | | | | | X | | | | | X | X | | | |
| Verify/Confirm | | | | X | X | | | | | | X | | X | |

[1] The gray shaded applications are MITRE applications, and the others are from DLR. An "X" indicates, which command semantics is used in which application.

### 3.3.4. Approach Clearance Usage Analysis (ACUA)

MITRE's voice data analytics capability was used to mine radio communications for approach clearances to inform a post-operational approach procedure utilization and conformance study [18]. The study used spoken approach clearances and radar tracks to detect trends in when and where flights received their approach clearances, correlation between aircraft equipage and approach clearance, and the effect of weather conditions on procedure utilization. The study was also able to use detected approach clearances to differentiate aircraft flying visual approaches from aircraft flying Required Navigation

Performance (RNP) procedures and then analyze RNP procedure conformance. For more information on this application, please see [18] for details.

### 3.3.5. Prefilling Radar Labels for Vienna Approach (PRLA)

DLR and Austro Control performed a validation exercise with 12 ATCos in DLR's ATMOS (Air Traffic Management Operations Simulator) from September 2022 to November 2022. The validations compared ATCos' workloads and safety effects with and without ASRU support. The evaluated application was inputting spoken commands into the aircraft radar labels on the radar screen. Therefore, the number of missing and wrong radar label inputs with and without ASRU support was determined. The details can be found in "Automatic Speech Recognition and Understanding for Radar Label Maintenance Support Increases Safety and Reduces Air Traffic Controllers' Workload" of Helmke et al. presented at the 15th USA/Europe Air Traffic Management Research and Development Seminar (ATM2023) in, Savannah, GA, USA, 5–9 June 2023.

### 3.3.6. Electronic Flight Strip in Multiple Remote Tower Environment (MRT)

In multiple remote tower operations, controllers need to maintain electronic flight strips for a number of airports. The manual controller inputs can be replaced by automatic inputs when using ASRU support. In the HMI interaction modes for the Airport Tower project, the tower/ground controller had to simultaneously take care of three remote airports. Their responsibilities included entering flight status changes triggered by issued clearances, such as pushback from gate, taxi with taxiways, line-up, runway clearances, etc., with an electronic pen into the flight strip system. When ASRU support was active, the flight status changes were automatically recognized from the controller utterances, entered into the flight strip system, and highlighted for their review. If an automatically detected flight status change was not manually corrected by the controller within ten seconds of entry, the values were accepted by the system. The prototypic system was validated with ten controllers from Lithuania and Austria in 2022. More details can be found in the presentation "*Understanding Tower Controller Communication for Support in Air Traffic Control Displays*" given at the SESAR Innovation Days in Budapest in 2022 by Ohneiser et al.

### 3.3.7. Integration of ASRU with A-SMGCS for Apron Control at Frankfurt and Simulation Pilots in Lab Environment (SMGCS and SPA)

In June 2022, Frankfurt Airport (Fraport), together with DLR, ATRiCS Advanced Traffic Solutions GmbH, and Idiap performed validation trials with 15 apron controllers in Fraport's tower training environment under the STARFiSH project. An A-SMGCS (Advanced Surface Movement Guidance and Control System) was supplemented with ASRU to enable integration of recognized controller commands into the A-SMGCS planning process and simultaneously improve ASRU performance with the addition of context from A-SMGCS. Together with manual input from the ATCo, the A-SMGCS is able to detect potentially hazardous situations and alert the ATCo. The addition of ASRU reduces the burden on the ATCo to manually input issued clearances over the radio into A-SMGCS. Research results showed that up to one third of the working time of controllers is spent on these manual inputs, which is detrimental to overall efficiency because ATCos spend less time on the optimization of traffic flow. More details can be found in the presentation "*Apron Controller Support by Integration of Automatic Speech Recognition with an Advanced Surface Movement Guidance and Control System*" given at the SESAR Innovation Days in Budapest in 2022 by Kleinert et al. Table 8 contains two columns for this application. The column "SMGCS" corresponds to the support of the ATCo in this application, whereas the column "SPA" corresponds to the support of the simulation pilots by ASRU.

### 3.3.8. Workload Prediction for London Terminal Area (WLP)

Under the Highly Automatic Air Traffic Controller Working Position with Artificial Intelligence Integration (HAAWAII) project, DLR, together with NATS (the Air Navigation

Service Provider of the United Kingdom), University of Brno, and Idiap developed a tool that determines an ATCo's workload in real-time, based on input from ASRU. The radio communications between ATCos and pilots at London TMA, for Heathrow Approach, was analyzed. Length of utterances, frequency usage rate, number of greetings, and number of miscommunications (say again, etc.) were evaluated for this purpose [30]. Callsign information is of minor importance here.

### 3.3.9. Integration of ASRU and CPDLC (CPDLC)

Under the HAAWAII project, DLR, together with NATS and Isavia ANS evaluated the performance of ASRU and CPDLC integration. More details can be found in the deliverable D5.3 of the HAAWAII project "*Real Time Capability and Pre-Filling Evaluation Report*". In the future, ATCos and pilots will communicate their intentions via both data link, e.g., CPDLC (Controller Pilot Data Link Communication), and radio communications. In this envisioned state, ASRU and CPDLC are not competitors but complementary tools. Current CPDLC applications are expected to advance with the advent of data link with lower latency (LDACS). ASRU can reduce the number and complexity of mouse clicks required to create a CPDLC message.

### 3.3.10. Pilot Weather Reports (PWR)

MITRE performed a post-operational analysis on the quantity of weather-related pilot reports (PIREPs) that could be automatically detected and submitted as "synthetic PIREPs" by an ASRU-enabled capability [19]. One of the goals of this analysis was to see if synthetic PIREPs could supplement the manually submitted PIREPs present in the system today and better inform strategic and tactical planning of ATC operations throughout the US National Airspace System (NAS) while also easing the ATCo workload. This use case relied on the Callsign and Pilot Report semantic representations to generate a formatted synthetic PIREP. More details about the motivation, outcomes, and conclusions of this analysis can be found in [19].

### 3.3.11. Use of Visual Separation (VFR)

Pilot-to-pilot visual separation is an important component of NAS safety and efficiency because it allows aircraft to fly closer together with the pilot assuming responsibility for separation. However, determining whether pilots were maintaining visual separation can only be determined from the voice communications between ATCo and pilot. ASRU can be used to detect traffic advisories (when an ATCo points out traffic to a pilot), the pilot reporting the traffic in sight, and the instruction for pilots to "maintain visual separation" in post-operations analysis. This information is critical to understanding the safety of a given encounter between aircraft. The information can therefore be used to better prioritize operations for safety assurance review. Visual separation information can also be used to inform efficiency-perspective analysis of operations (e.g., what percentage of flights are visual separated), because it informs the spacing between aircraft, which informs throughput/capacity.

### 3.3.12. Simulation Pilots in Enroute Domain Controller Training (SPET)

MITRE designed and prototyped high-fidelity simulation training consoles to support controller training in the enroute domain [5]. To reduce training and simulation costs, these consoles included a real-time simulation pilot system that uses automatic flight management, ASRU, and text-to-speech technology to interact with controllers during training simulations. Automated simulation pilots can handle more aircraft workload, provide consistent performance and response times to controller instructions, and require less training than human simulation pilots. The success of this prototype led to other follow-up projects, such as terminal training applications, Human-In-The-Loop (HITL) simulations to support new technology prototyping, procedure and airspace design, and

research studies in MITRE's Integration Demonstration and Experimentation (IDEA) Lab, and bilingual training consoles for international use.

### 3.3.13. Readback Error Detection for Enroute Controllers (RB-E)

Under the HAAWAII project, DLR, together with the Icelandic Air Navigation Service Provider Isavia ANS, University of Brno, and Idiap developed a readback error detection assistant and tested it on pilot and ATCo voice utterances directly recorded in the ops room environment of Isavia ANS [2].

### 3.3.14. Readback Error Detection for Tower Controllers (RB-T)

In 2016, MITRE conducted a feasibility study into the automatic detection of readback errors at the tower/local controller ATCo position using recorded live-operations audio [1]. The study focused on runway and taxiway use clearances and assessed the readiness of ASRU performance to support this type of application. Whereas automatic speech recognition performance was promising, the study found that more complex understanding logic was needed to differentiate acceptable readback discrepancies from alert-worthy readback errors. The study also identified the importance of detecting the nuances of dialogue between the ATCo and pilot during which the ATCo might have already taken corrective action and nullified the need for an alert.

### 3.4. Application-Specific Metrics

We previously described general semantic accuracy metrics for evaluating how well labeled concepts are extracted in general, irrespective of a downstream application. In an ideal world, we could have a single set of objective ASRU metrics that could be used to communicate accuracy and be meaningful across all applications. However, we cautioned that these general semantic metrics should be supplemented before use with a downstream application. In this section, we describe why metrics must be tailored to the application in order for it to be useful.

The first set of metrics to consider is the set that describes the accuracy performance of the application, i.e., the performance that is relevant to the end user (who could be an ATCo, pilot, data analyst, policy maker, etc.). The application accuracy is the ultimate measure of performance because the application's benefit is the ultimate measure of the utility of the capability.

However, there are situations where the application accuracy can diverge from the accuracy of the underlying ASRU. One case is when the application logic is such that an incorrect ASRU result can still produce the correct application output. Another case is when there is non-speech information used after ASRU processing that can improve wrong or missing ASRU output.

For example, consider the application described in Section 3.3.3, in which ASRU is used to detect the ATCo landing clearance and then surveillance track information is used to determine if the arrival is lined up for the correct runway. If the arrival is lined up for the wrong runway, the application issues an alert to the ATCo; if no landing clearance is detected for an arrival, the application does nothing.

Incorrect ASRU detection of the callsign will likely result in no alert because the system will not be able to compare the flight's track with a clearance. No alert will likely be the correct application response because most arrivals line up for the correct runway. Similarly, missing the landing clearance would also result in no alert. In other words, we are getting the right results but for the wrong reason.

In contrast, incorrect ASRU detection of the callsign could be corrected through use of other information, e.g., using the arrival's position in the landing sequence to fill in the gap in knowledge, resulting in correct application performance.

It is clear from these examples that although application performance is the ultimate measure of success, it obscures some detail of the ARSU accuracy. Detail of the ASRU accuracy can be critical for two reasons. One, it provides understanding of what kinds of

application errors will result from ASRU errors. Two, it provides understanding of where ASRU accuracy can and should be improved.

Continuing the example of using ASRU to detect landing clearances that can be compared with arrival alignment to identify wrong surface alignment, ASRU errors in callsign recognition will result in ASRU failing to associate the landing clearance with the correct aircraft. Given that most aircraft line up correctly, this missed recognition will likely still result in a correct "no alert" response at the application level. On the other hand, ASRU errors in runway recognition could result in ASRU producing an incorrect assigned runway for the flight, which could then result in a false alert to the ATCo.

Thus, for an application that aims to detect and alert on runway misalignment, the ASRU accuracy measures should be defined corresponding to the ontology concepts that need to be detected for the application: callsign, landing clearance, and runway. For each concept, detection accuracy can be evaluated using the metrics defined in Table 5.

These metrics should then be produced for each concept separately, such that callsign, landing clearance, and runway would each have several associated accuracy measures: recall, precision, etc. These metrics can then be used to identify and measure performance improvements in the ASRU. For example, they differentiate between missed landing clearances due to missed callsign detection and those due to missed landing clearance detection.

Note that the concept detection accuracy can be rolled up into a single metric, producing an overall concept recognition error rate by combining the TP, FP, TN, and FN for all concepts. This overall concept recognition error rate provides a general measure of the ASRU accuracy, and improvement in this measure generally means better ASRU accuracy for the application, which in turn means better overall efficacy for the application. However, as the previous examples illustrate, rolling the detection of these concepts up into a single measure will obscure understanding about the effects of the errors on application performance or where ASRU improvements should be targeted. Using Figure 5, consider the following example.

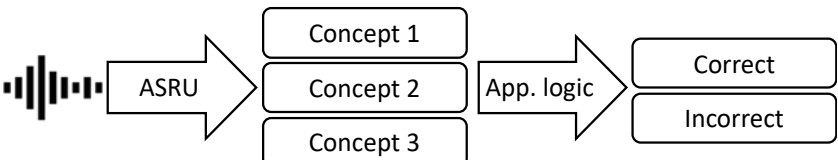

**Figure 5.** Example use of ASRU semantic concepts for a specific application.

Consider evaluation of ASRU performance on a set of 10 transmissions for this hypothetical application where all three basic concepts are needed to generate correct application output. A concept can be the callsign, the command type, the command value, etc. The Concept Error Rate (CER) measures the accuracy of the ASRU in detecting each concept, and a CER *should* be measured for each concept, not combined into a single metric covering the accuracy of detecting all semantic concepts. In contrast, the Command Recognition Error Rate (CRER), as defined in Table 5, measures the accuracy of the ASRU in detecting complete commands, which requires both the callsign and the instructions, which can be composed of different concepts, again.

In Case A, ASRU produces fully correct concepts for nine of the ten transmissions but zero correct concepts for one transmission. A "combined" concept error rate (CER, 3/30 = 10%) and the application error rate (1/10 = 10%) are the same. In Case B, ASRU produces fully correct output for seven of the ten transmissions but two out of three correct concepts for the remaining three transmissions. The combined CER is still 3/30 = 10% but the application error rate is now 3/10 = 30%. The CRER for Case A is 10% whereas the CRER for Case B is 30%.

The application performance for Case A is clearly better than for Case B. It is clear from this example that combined CER is obscuring important information. First, Case A will result in better application performance than Case B, despite the two having the same combined CER. Second, neither the combined CER nor the CRER tells us which concepts

have room for improvement. For the example in Case A, the issue may be a systematic problem with a transmission that affects the recognition of all three concepts, such as bad audio or incorrect segmentation. For the example in Case B, did the system miss the callsign each time or one of the other concepts? Individual measures of precision and recall for each ontology concept (callsign, landing clearance, and runway in the example used above) are needed to fully assess the ASRU accuracy.

As another example, if the application only requires one concept to be detected (e.g., the closed runway operation clearance detection application described in Section 3.3.2) and does not require a callsign, then a metric such as CRER is not appropriate because it incorporates unnecessary concepts into the metric.

In summary, there is not a single metric nor type of metric that is appropriate for all applications. Practitioners should develop metrics specific to the application, covering both the application level (i.e., the performance of the application from the user's perspective) and the ASRU level (i.e., the performance of the ASRU on individual concepts needed for the application). These application-specific metrics may expand beyond accuracy measures and incorporate requirements on computing and speed performance as applications come closer to being fielded in operational settings with specific resource constraints and demands on response time.

## 4. Quantitative Analyses with Applied Ontologies

Thus, application-specific metrics assess overall application readiness for an operational setting and acceptability to the end-user. In this capacity, they are as important, if not more so, than the lexical and semantic level ontology when applications are transplanted into new operational environments. The general semantic accuracy metrics we described previously help researchers evaluate data, algorithms, and models; however, application-specific metrics describe the end-user experience and how he or she will be impacted by the addition of the application to the operational environment. For this reason, we recommend application-specific metrics be added to the conceptual-level definitions and rules of the ontology when an application is transitioned. These application-specific metrics can go beyond TN/TP/FN/FP and include metrics even more relevant to operations, such as false alerts per hour.

The following two subsections describe example applications and the types of ontology-related metrics needed to assess their accuracy performance.

### 4.1. Application-Specific Metrics for a Workload Assessment in the Lab Environment

This application is briefly described in Section 3.3.5. Table 9 summarizes the applicable semantic concepts relevant to this application. The impact on workload and safety was measured in terms of the number of missing and incorrect radar label inputs when ASRU support was present and when it was not.

**Table 9.** Semantic accuracy metrics for workload assessment.

| WER | Total | TP | FP | FN | TN | RcR | RER | RjR | Prc | Rec | Acc | F-1 | F-2 | F-0.5 |
|---|---|---|---|---|---|---|---|---|---|---|---|---|---|---|
| 0.0% | 17,096 | 16,933 | 71 | 94 | 11 | 99.1% | 0.4% | 0.5% | 99.6% | 99.4% | 99.0% | 99.5% | 99.5% | 99.6% |
| 3.1% | 17,096 | 15,869 | 368 | 920 | 10 | 92.9% | 2.2% | 5.4% | 97.7% | 94.5% | 92.5% | 96.1% | 95.1% | 97.1% |

Table 9 summarizes the command detection accuracy when ASRU support was present during operations. Row "0.0%" shows the command detection performance with a perfect speech-to-text conversion, i.e., all incorrect detections come from errors in semantic extraction. Row "3.1%" shows the actual command detection performance during the validation trials with a speech-to-text engine that had an average WER of 3.1%.

For this use case, the application-specific metrics closely aligned with the semantic accuracy metrics described in Section 3.1.2 because the command detection accuracy translated directly into radar label entry accuracy. The number of correctly detected commands, or the command recognition rate (RcR), translated into how many entries the ATCo did not have to manually enter into the automation system. The number of incorrectly detected

commands, or the command recognition error rate (CRER), translated into the number of safety risks introduced due to incorrect radar label inputs. The metric recall corresponded approximately to the command detection accuracy. They would be equal if TP+FP+FN+TN was equal to the total number of command samples (Total). The metric RER approximated 1—Prc. This correlation between RcR and Acc and the inverse correlation between RER and Prc was not present in our nominal example in Section 3.1.2 but was present in this experiment.

### 4.2. Application-Specific Metrics for a Post-Operations Pilot Report Analysis

The application itself is briefly described in Section 3.3.10. For the context of this paper, we discuss here the value of the application-level metrics used to measure the validity of this prototyped application's overall performance.

From the analyst perspective, the relevant metrics for this application were:

1. The number of correctly detected and accurately formatted pilot reports (PIREPs), i.e., correct PIREPs.
2. The number of correctly detected but incorrectly formatted PIREPs (incorrect PIREPs because they are incomplete, misleading, or both).
3. The number of PIREPs not detected or not mapped to a formatted PIREP (missed PIREPs).

The first quantity informs how much reliable supplemental information could be introduced into the US National Airspace System (NAS) by this capability. The second quantity informs how much supplemental information introduced might be misleading and potentially detrimental to planning. The final quantity informs how much potential supplemental information is being missed but would not negatively affect planning except by omission.

However, there is not a direct one-to-one correspondence between the semantic accuracy of the individual Callsign and Pilot Report concepts and the application metrics. Figure 6 illustrates the effect of different errors during the automatic PIREP detection logic within the application and their effect on the overall application performance. As the diagram shows, an error in Callsign extraction could lead to either an incorrect PIREP or a missed PIREP; an error in Pilot Report extraction could also independently lead to an incorrect PIREP or a missed PIREP, and only the combined accurate extraction of both the Callsign and Pilot Report semantics could lead to a correct PIREP.

Table 10 recaps the concept metrics of the application originally published in [19]. The final output quantities show that even when a PIREP concept is correctly detected, it may not be fully and correctly encoded (i.e., the application-level success).

Using the sample results from Table 10, we define application-specific metrics for precision and recall. We define true positive PIREPs as those that are encoded with complete information *and* PIREPs that are encoded with correct but incomplete information, on the reasoning that some information is better than none; this is an application-specific consideration. Using that definition, we calculate precision as 88% = (79 + 26)/(79 + 26 + 14). Recall is then calculated as 63% = (79 + 26)/168.

The complexity of the final application metrics is compounded by additional upstream probabilistic processes such as speech diarization, speech recognition, and text classification that could all introduce errors affecting the final result of the application. The interwoven effects of the different internal model and algorithm errors mean that no one model or algorithm is the most important and no individual model or algorithm accuracy metric could estimate overall application accuracy. Thus, the application-specific metrics are necessities invaluable for assessing the overall value of the prototype and its readiness for use in an operational setting.

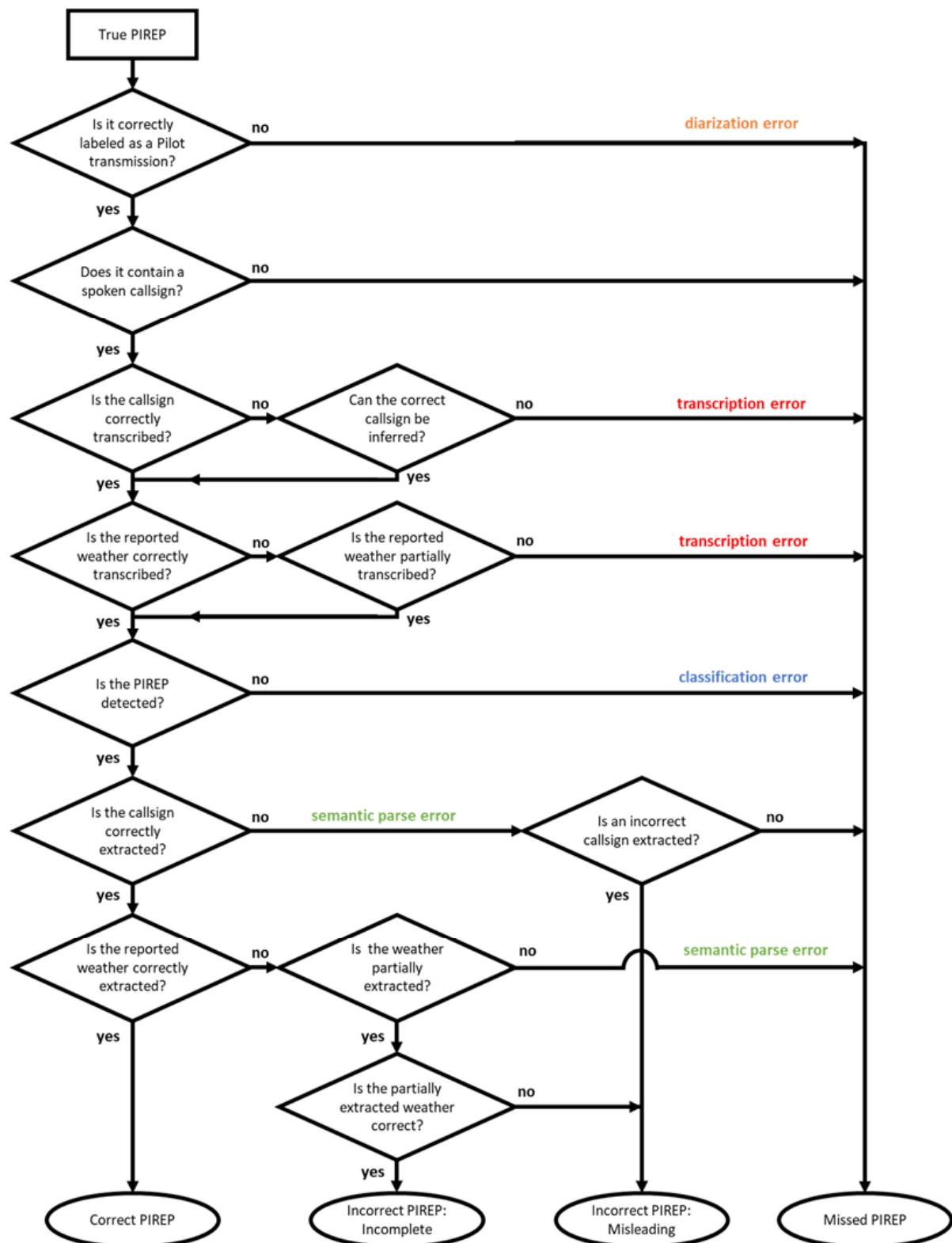

**Figure 6.** Effect of different ASRU errors on final PIREP application performance.

**Table 10.** Summary of PIREP application accuracy metrics.

| Ground Truth | | Detection | | Encoding | |
|---|---|---|---|---|---|
| 168 | PIREP | 161 | Correct detection | 79 | Correct final PIREP |
| | | | | 34 | Incorrect discard—callsign not spoken |
| | | | | 8 | Incorrect discard—callsign not detected |
| | | | | 26 | Incorrect final PIREP—missed details |
| | | | | 14 | Incorrect final PIREP—incorrect flight |
| | | 7 | Missed detection | | |
| 96 | Not PIREP | 79 | Correct rejection | | |
| | | 17 | False detection | | |

### 4.3. European Word-Level Challenges and Statistics

As already described in Section 3.2.1, a lot of the vocabulary that appears in ATCo–pilot communications are general-purpose words such as climb, descend, cleared, etc. A large and significant percentage of the vocabulary is made up of names, e.g., airline designators, facility identifiers, location identifiers, navigational aids, and procedure identifiers. They, however, seldom occur, i.e., training data might not be available as needed.

The following Table 11 shows the results of the top 10 words in the two applications from the laboratory environment, described in Sections 3.3.5 and 3.3.7. "# Spoken" shows how often the word was really said. "Freq" shows how often this word was recognized relative to the number of all words spoken.

**Table 11.** Top 10 words of Vienna Approach and Frankfurt Apron Control.

| Vienna Approach | | | Frankfurt Apron Control | | |
|---|---|---|---|---|---|
| **Word** | **# Spoken** | **Freq** | **Word** | **# Spoken** | **Freq** |
| two | 8841 | 7.4% | one | 11,724 | 9.3% |
| one | 8128 | 6.8% | november | 7713 | 6.1% |
| zero | 7576 | 6.4% | five | 7100 | 5.6% |
| four | 5805 | 4.9% | two | 5520 | 4.4% |
| three | 5624 | 4.7% | lufthansa | 4994 | 4.0% |
| eight | 5422 | 4.6% | eight | 4939 | 3.9% |
| austrian | 4979 | 4.2% | lima | 4002 | 3.2% |
| six | 4295 | 3.6% | seven | 3882 | 3.1% |
| seven | 4028 | 3.4% | four | 3769 | 3.0% |
| descend | 3909 | 3.3% | hold | 3513 | 2.8% |

Words shaded by "light blue" were not present both top 10 lists.

The Vienna data is based on 118,800 spoken words, whereas the Apron application is based on 125,800 spoken words. In "light blue", we marked the words that were only present in one or the other top 10 list but not in both. In Frankfurt, most of the taxi way names start with the letter "N", e.g., N1, N6, etc. Most of the flights to and from Vienna are from "Austrian Airlines", whereas it is "Lufthansa" for Frankfurt.

Table 12 shows the top 10 word for London TMA (Section 3.3.8) and for the enroute traffic managed by Isavia ANS (Section 3.3.9). The fact that "Reykjavik" is within the Top 10 of Icelandic traffic control is quite clear. Reykjavik is the capital of Iceland and the station name ATCos and pilots are using. "speed" being the sixth most frequent used word in London traffic might be surprising; however, knowing that "speed" is used both in speed commands and also in the callsign "speed bird" (for British Airways) explains the high occurrence. The London data is based on 102,952 spoken words, whereas the enroute application is based on 73,980 spoken words.

**Table 12.** Top 10 words of London TMA and Isavia Enroute Traffic.

| London TMA | | | Isavia ANS Enroute Traffic | | |
|---|---|---|---|---|---|
| **Word** | **# Spoken** | **Freq** | **Word** | **# Spoken** | **Freq** |
| one | 7599 | 7.4% | one | 4371 | 5.9% |
| zero | 6284 | 6.1% | zero | 3849 | 5.2% |
| five | 5191 | 5.0% | three | 3255 | 4.4% |
| two | 5019 | 4.9% | five | 3230 | 4.4% |
| seven | 3702 | 3.6% | seven | 3064 | 4.1% |
| speed | 3677 | 3.6% | two | 2830 | 3.8% |
| three | 3536 | 3.4% | six | 2436 | 3.3% |
| six | 3198 | 3.1% | reykjavik | 2202 | 3.0% |
| four | 3113 | 3.0% | nine | 2057 | 2.8% |
| eight | 2965 | 2.9% | four | 1962 | 2.7% |

Words shaded by "light blue" were not present both top 10 lists.

Investigating the statistics for all four ASRU applications, we get the values shown in Table 13. The ten digits make the top 10. The digit "four" has the highest word error rate. It is often mixed with "for", a problem which can be solved afterwards at the semantic level.

**Table 13.** Top 10 words of four DLR applications from Tables 11 and 12.

| Word | # Spoken | Freq |
|---|---|---|
| one | 31,822 | 7.6% |
| two | 22,210 | 5.3% |
| zero | 19,378 | 4.6% |
| five | 19,266 | 4.6% |
| three | 15,346 | 3.7% |
| eight | 15,085 | 3.6% |
| seven | 14,676 | 3.5% |
| four | 14,649 | 3.5% |
| six | 13,313 | 3.2% |
| nine | 9998 | 2.4% |

Table 14 shows the "Number of Words" evaluated for each of the four applications. For Vienna, 179 words were observed more than four times, i.e., at least five times. The first 62 most occurring words for Vienna already sum up to 95% of all the spoken words. For 99% of all spoken words, we need 112 words. All in all, we have 347 different words observed for the Vienna ASRU applications (row "words for 100%").

**Table 14.** Statistics on word level for different ASRU applications.

| | Vienna | Frankfurt | NATS | Isavia | All |
|---|---|---|---|---|---|
| Number of Words | 118,794 | 125,810 | 102,952 | 73,980 | 421,536 |
| Spoken >4 times | 179 | 291 | 497 | 583 | 931 |
| Words for 95% | 62 | 110 | 205 | 322 | 256 |
| Words for 99% | 112 | 203 | 432 | 754 | 619 |
| Words for 100% | 347 | 520 | 899 | 1375 | 1972 |

The word statistics in Table 14 also show the difference between lab experiments and real-life data from the ops room. The number of used words is much bigger in the ops room environment than in the lab environment. This is supported by the number of words occurring more than four times and also by the 95%, 99%, and 100% thresholds. In the Icelandic enroute airspace English, Icelandic and Norwegian words are used, which explains the high number of different words.

### 4.4. US Word-Level Statistics

A similar analysis has been performed by MITRE. It is based on 70 ATC facilities all over the US with a corpus of 1,248,436 words. Table 15 is similar to Table 14 for the European word-level statistics.

**Table 15.** Statistics on word level for different MITRE data sets.

| From Corpus Partition of 99,513 Transmissions/1,248,436 Words | | | |
|---|---|---|---|
| Unique Words | | Cumulative Word Count Percentage | |
| Spoken >1 time | 4471 | 1st 50 words | 60% |
| Spoken >4 times | 2640 | 1st 100 words | 74% |
| Words for 95% | 542 | 1st 150 words | 81% |
| Words for 99% | 1884 | 1st 500 words | 94% |
| Words for 100% | 7236 | 1st 1000 words | 98% |

Table 16 shows the top 10 word occurrences from the MITRE analysis. The 10 digits are also the most frequently used words in the US.

**Table 16.** Occurrence of digits in MITRE data set.

| Word | # Spoken | Freq | Additional Information |
|---|---|---|---|
| one | 56,298 | 4.5% | |
| two | 54,376 | 4.4% | |
| three, tree | 45,112 | 3.6% | tree: 167 |
| zero, oh | 43,584 | 3.5% | oh: 3168 |
| five, fife | 32,038 | 2.6% | fife: 1 |
| four | 31,035 | 2.5% | |
| seven | 27,466 | 2.2% | |
| six | 26,410 | 2.1% | |
| eight | 22,324 | 1.8% | |
| nine, niner | 21,901 | 1.8% | niner: 7193 |
| All | 360,544 | 29.0% | |

Looking into the details we observe some other interesting differences such as "one" and "two" also being the top words in US. The word "nine" would be rank only sixteen by occurrence; however, when combined with "niner", the composite moves into the top 10 in terms of occurrence frequency. One surprising observation is that "nine" is used more often than "niner", although niner is the recommended spoken form for the digit by the ICAO [28]. The European transcription ontology does not even distinguish between "nine" and "niner". Both words are mapped to "nine". Europe also does not distinguish between "five" and "fife" or "three" and "tree". Manual transcribers may not have even been able to distinguish between them.

The digit "oh" for "zero", transcribed in Europe as a capital "O", is observed in the European data only 59 times and only in the operational environment data sets from NATS and Isavia. This is a negligible percentage. However, in the US data, the more than 7000 occurrences constitute a significant percentage.

The 10 digits from "zero" to "nine" cover 42% of all words observed in the European DLR data set. In the MITRE data set, the same digits comprise 29% of all spoken words, when "niner", etc., are also considered. Our hypothesis for this is that ATCos and pilots are not limited to the ten digits, as recommended by ICAO [28]. They also use the other group-form digit words such as "ten", "twenty", "thirteen", "fourteen", "hundred", "thousand", etc. When these additional numbers are summed up together with "zero" through "nine", then numerical words comprise 40% of all words spoken, which is shown in Table 17.

**Table 17.** Frequency of values between 10 and 1000 in MITRE and DLR data sets.

| Word | Numerical Value | MITRE | | DLR | |
| --- | --- | --- | --- | --- | --- |
| | | # Spoken | Freq | # Spoken | Freq |
| ten | 10 | 4033 | 0.3% | 270 | 0.1% |
| eleven | 11 | 2788 | 0.2% | 8 | 0.0% |
| twelve | 12 | 3185 | 0.3% | 5 | 0.0% |
| thirteen | 13 | 1810 | 0.1% | 2 | 0.0% |
| fourteen | 14 | 2274 | 0.2% | 4 | 0.0% |
| fifteen | 15 | 2671 | 0.2% | 13 | 0.0% |
| sixteen | 16 | 2224 | 0.2% | 5 | 0.0% |
| seventeen | 17 | 2085 | 0.2% | 3 | 0.0% |
| eighteen | 18 | 2251 | 0.2% | 62 | 0.0% |
| nineteen | 19 | 2404 | 0.2% | 327 | 0.1% |
| twenty | 20 | 17,323 | 1.4% | 972 | 0.2% |
| thirty | 30 | 14,773 | 1.2% | 101 | 0.0% |
| forty | 40 | 11,961 | 1.0% | 45 | 0.0% |
| fifty | 50 | 11,327 | 0.9% | 201 | 0.0% |
| sixty | 60 | 7907 | 0.6% | 286 | 0.1% |
| seventy | 70 | 6882 | 0.6% | 14 | 0.0% |
| eighty | 80 | 7339 | 0.6% | 317 | 0.1% |
| ninety | 90 | 6401 | 0.5% | 21 | 0.0% |
| hundred | 100 | 4726 | 0.4% | 1329 | 0.3% |
| thousand | 1000 | 13,732 | 1.1% | 5019 | 1.2% |
| All | | 128,096 | 10.3% | 9004 | 2.1% |

The words "hundred" and "thousand" have nearly the same frequency in the MITRE and DLR data sets. These words are recommended by ICAO. The combined occurrences of words for 11 through 90 are negligible in DLR's data set. They sum up to only 0.6% of the words spoken, whereas in the MITRE data sets they sum up to over 10%, which is significantly more. Furthermore, analysis of the US data set by speaker showed that ATCos and pilots used group-form numbers about equally, so the difference in group-form word occurrence between the US and European data sets can be attributed to differences in word usage by region, i.e., between the US and Europe, not speaker.

Moreover, very interesting is how small a percentage of the most frequently occurring words in the data set comprise in the overall data set vocabulary. Table 18 summarizes the top occurring words that comprise 95% of words in the data set and the percentage of the vocabulary they represent. This top 95% of words present in the corpus is made of 551 distinct words and includes all the numbers and letters but not most of the airline, ATC facility, and waypoint names. This 551-word set is about 7.61% of the data set's 7236 distinct word vocabulary, which means the remaining 92.39% of the distinct words in the vocabulary comprise only 5% of the data corpus in terms of occurrence.

This last statistic illustrates one of the biggest challenges for ASRU in the ATC domain. The large variety of distinct waypoint, airline, and airport names relevant to understanding is hard to recognize correctly because they have low occurrence in the data set. The reason for their low occurrence is because a training corpus for ASR or semantic parse is often deliberately varied to improve robustness and reduce overfitting, which means they are collected from many facilities and regions. However, the geographical spread of the audio data sources, while improving general robustness, dilutes the observation frequency of regional waypoint, airline, and facility names. This scarcity of a large percentage of the vocabulary in the training data subsequently leads to misrecognition of these words and misinterpretation unless deliberate action is taken to correct or improve their detection.

The findings of this analysis lead to our conclusion that although the methods and tools for developing and measuring ASRU performance can be shared across regions (e.g., between the US and Europe), the specific models built for specific regions would likely not work well across regions.

**Table 18.** Word classification of MITRE words.

| Meaning Category | Definition | Examples | # Spoken | Percentage of Corpus Words | Percentage of Vocabulary |
|---|---|---|---|---|---|
| Other | | climb, fly, contact, thanks, until | 527,579 | 42.3% | 4.78% |
| Numeric | Digits, other numbers, number modifiers | zero, ten, hundred, triple, point | 498,066 | 39.9% | 0.51% |
| Callsign Words | Airline names, aircraft types, air service types | United, Cessna, Medevac | 56,265 | 4.5% | 0.90% |
| Phonetic Alphabet | Phonetic alphabet words | Bravo, Charlie, Zulu | 48,077 | 3.9% | 0.39% |
| Place Names | ATC facilities and airport names | Atlanta, Reno | 21,958 | 1.8% | 0.50% |
| Initials | Letters, e.g., "i l s" | V, O, R, J, F, K, D, F | 16,543 | 1.3% | 0.35% |
| Filler Words | Words that fill up space but do not add substance | uh, um | 10,356 | 0.8% | 0.03% |
| Multiple Meanings | Words that can be all or part of airline names, airport names, or general-purpose words | Sky, Midway, Wisconsin | 7084 | 0.6% | 0.12% |
| Waypoint Names | Named fixes and waypoints | SAILZ, KEEEL, HUNTR, KARLA | 706 | 0.1% | 0.04% |
| Total | | | 1,186,634 | 95.0% | 7.61% |

## 5. Conclusions

This paper built off our comparative analysis of the two ontologies in [3] in two ways. First, this paper describes the impact of ontologies on collaboration on data, models, and applications. We described several ways that an ATC ontology is critical to facilitating collaboration between researchers and to appropriate evaluating ASRU applications in the ATM domain, using examples of specific applications to illustrate how ontology facilitates development of the metrics targeted for the application.

Second, this paper presents a word-level comparison of US and European ATC speech, specifically focusing on similarities and differences in the types of words. Although there are significant similarities (e.g., in both regions, digits make up the top 10 most spoken words), there are also significant differences (e.g., the frequency of group-form numbers). This analysis leads to our conclusion that whereas the methods and tools for developing and measuring ASRU performance can be shared across regions (e.g., between US and Europe), the specific models built for the different regions would likely not work well across regions.

Future work is needed to develop capabilities to make methods and tools more shareable between ontologies. This effort could involve modifying one or both ontologies and/or creating translation mechanisms to automatically convert data from one ontology to the other. Ultimately, research funding is critical to informing the effective and available paths forward.

**Author Contributions:** Conceptualization, S.C., H.K., R.M.T. and H.H.; data curation, S.C., H.K. and M.K.; formal analysis, H.H., S.C. and R.M.T.; funding acquisition, H.K. and H.H.; investigation, S.C., R.M.T., H.K. and H.H.; methodology, H.H., R.M.T. and S.C.; project administration, H.H. and H.K.; resources, H.H., O.O. and H.K.; software, H.H., S.C. and M.K.; supervision, H.K., R.M.T. and H.H.; validation, R.M.T.; visualization, S.C. and O.O.; writing—original draft, S.C., H.K. and R.M.T.; writing—review and editing: O.O., H.H., R.M.T., H.K., S.C. and M.K. All authors have read and agreed to the published version of the manuscript.

**Funding:** This is the copyright work of The MITRE Corporation, and was produced for the U. S. Government under Contract Number 693KA8-22-C-00001, and is subject to Federal Aviation Administration Acquisition Management System Clause 3.5-13, Rights In Data-General, Alt. III and Alt. IV

(Jan 2009). No other use other than that granted to the U. S. Government, or to those acting on behalf of the U. S. Government, under that Clause is authorized without the express written permission of The MITRE Corporation. For further information, please contact The MITRE Corporation, Contracts Management Office, 7515 Colshire Drive, McLean, VA 22102-7539, (703) 983-6000. Approved for Public Release; Distribution Unlimited. Public Release Case Number 23-1072. DLR acknowledges the support of the SESAR Joint Undertaking for developing the first version of the ontology in PJ.16-04-W2 SESAR2020 (grant agreement No 734141) and for funding the extended version in the HAAWAII project (grant agreement No 884287).

**Data Availability Statement:** Data can be made available by contacting the corresponding author if the data is not protected by, e.g., GDPR or other contracts.

**Acknowledgments:** Many thanks to Yuan-Jun Wei and Weiye Ma for their curation of the MITRE ASR corpus. MITRE also acknowledges the support from the FAA for the research funding and for allowing access to recorded ATCo–pilot live-operations radio voice communications.

**Conflicts of Interest:** The authors declare no conflict of interest. The funders had no role in the design of the study; in the collection, analyses, or interpretation of data; in the writing of the manuscript; or in the decision to publish the results.

## Appendix A. Command Types in European and MITRE Ontology

**Table A1.** Altitude clearances in MITRE and European ontology.

| MITRE | European | Example/Explanation |
|---|---|---|
| Climb | CLIMB | climb to flight level three two zero |
| Descend | DESCEND | descend to flight level one four zero |
| Tries always to derive, whether CLIMB or DESCEND | ALTITUDE | if no descend or climb keyword is provided/recognized in transmission |
| StopAltitude | STOP_ALTITUDE/ STOP_CLIMB/ STOP_DESCEND | stop descent at flight level one zero zero |
| Maintain | MAINTAIN ALTITUDE/ PRESENT_ALTITUDE | maintain flight level one eight zero; maintain present level |
| Cancel | NO_ALTI_RESTRICTIONS | No altitude constraints at all. |

**Table A2.** Speed clearances in MITRE and European ontology.

| MITRE | European | Example/Explanation |
|---|---|---|
| IncreaseSpeed | INCREASE/INCREASE_BY | increase to zero point eight four mach |
| ReduceSpeed | REDUCE/REDUCE_BY | reduce speed to two two zero knots |
| Tries always to derive, whether REDUCE or INCREASE. | SPEED | if no reduce or increase keyword is provided/recognized in transmission |
| | RESUME_NORMAL_SPEED | Still the published speed constraints are relevant. |
| Cancel SpeedRestriction | NO_SPEED_RESTRICTIONS | The speed restriction is removed |
| DoNotExceed | OR_LESS used as qualifier | Speed limit |
| Maintain | MAINTAIN SPEED/PRESENT_SPEED | maintain present speed |
| SpeedChange | | SPEED, INCREASE, REDUCE used in Europe |
| | REDUCE_FINAL_APPROACH_SPEED | reduce final approach speed |
| | REDUCE_MIN_APPROACH_SPEED | reduce minimum approach speed |
| | REDUCE_MIN_CLEAN_SPEED | reduce minimum clean speed |
| | HIGH_SPEED_APPROVED | speed is yours |

**Table A3.** Altitude change rate clearances in MITRE and European ontology.

| MITRE | European | Example/Explanation |
|---|---|---|
| Climb (At) | RATE_OF_CLIMB | climb with two thousand feet per minute (or greater)/ climb at three thousand feet per minute |
| Descend (At) | RATE_OF_DESCENT | descend with two thousand five hundred feet per minute |
| Maintain | | maintain three thousand in the climb/ maintain three thousand five feet per minute in the climb |
| | VERTICAL_RATE | if no climb or descent keyword is provided/recognized in transmission |
| | EXPEDITE_PASSING | expedite passing flight level three four zero |

**Table A4.** Heading clearances in MITRE and European ontology.

| MITRE | European | Example/Explanation |
|---|---|---|
| TurnLeft, TurnRight | HEADING/TURN/TURN_BY (Qualifier LEFT/RIGHT) | turn left heading two seven zero; turn right by one zero degrees |
| Turn | TURN/TURN_BY (Qualifier LEFT/RIGHT) | turn right by one zero degrees |
|  | TURN (without a value) Qualifier LEFT/RIGHT) | turn right |
| Fly | HEADING (Qualifier none) | fly heading three six zero (no keyword left/right recognized) |
| Maintain | CONTINUE_PRESENT_HEADING/MAINTAIN HEADING | continue present heading |
|  | MAGNETIC_TRACK | magnetic track one one five |

**Table A5.** Routing clearances in MITRE and European ontology.

| MITRE | European | Example/Explanation |
|---|---|---|
| Direct | DIRECT_TO/DIRECT Approach_Leg/LatLong | direct to delta lima four five five/ direct final runway three four direct six zero north zero one five west |
| Resume | NAVIGATION_OWN | own navigation |
| Cleared | CLEARED TO | cleared to london heathrow |
| Circle | ORBIT (Qualifier LEFT/RIGHT) | make orbits to the left |

**Table A6.** Procedure clearances in MITRE and European ontology.

| MITRE | European | Example/Explanation |
|---|---|---|
| Cleared (STAR/SID/Approach) | CLEARED/CLEARED VIA/MISS_APP_PROC | cleared via sorok one november |
| Intercept (Approach/ApproachType) | INTERCEPT_LOCALIZER | intercept localizer for runway |
|  | INTERCEPT_GLIDEPATH | intercept glidepath |
|  | JOIN_TRAFFIC_CIRCUIT | right traffic circuit for runway three four |
| Join | TRANSITION | join nerdu four november transition |
| Resume | NAVIGATION_OWN | resume navigation |
| Continue | CONTINUE_APPROACH | continue approach runway zero one |
| Cleared | CLEARED Approach_Type | cleared Rnav approacch zero nine center |
| Cancel (Approach/ SpeedRestriction/ AltitudeRestriction) | CANCEL Approach_Type | cancel approach for runway zero five |
| Climb (Via) |  | climb via the capital one departure |
| Descend (Via) |  | descend via the cavalier four arrival |
|  | GO_AROUND | go around |

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
