# Peer review of "Effects of Language Ontology on Transatlantic Automatic Speech Understanding Research Collaboration in the Air Traffic Management Domain"

_aerospace, doi:10.3390/aerospace10060526_

Round 1

Reviewer 1 Report

This paper examines Automatic Speech Recognition and Understanding (ASRU) in the Air Traffic Management (ATM) domain.  It describes ways in which ontologies facilitate the sharing of data, models, algorithms, metrics, and applications.  It does comparative analysis of word frequencies in ATC speech in the United States and Europe.

This is clearly an application paper, as there are no new ASRU methods presented.  The paper is very well written and organized; it was a pleasure to read.  The problem area is of clear importance. I recommend acceptance.

Specific points:

..each exchanged 100 transmissions, .. - why such a small number? I would suggest a larger dataset.

..require special handling and .. ->

..requires special handling and ..

..that 551words cover .. ->

..that 551 words cover ..

..of words, i.e. the .. ->

..of words, i.e., the ..

..facilities! ->

..facilities.   (avoid use of exclamation points..rarely a good idea)

..Automatic Speech recognition and understanding (ASRU) .. - this acronym is used often; no need to redefine it in the text (line 522)

..determines ATCo’s workload .. ->

..determines an ATCo’s workload ..

..and CPDLC (CPDLC) .. - why repeat here?

.. future, ATCo and pilots ..

.. future, ATCos and pilots ..

..(when ATCo points .. ->

..(when an ATCo points ..

..determine, if the arrival .. ->

..determine if the arrival ..

..we got lucky! - not good to include in a technical paper

..nor the CRER tell us ..

..nor the CRER tells us ..

..marked the words, which were ..

..marked the words that were .. (comma makes a big difference)

..by e.g. GDPR ..

..by, e.g., GDPR ..

..to derive, whether it .. ->

..to derive whether it ..

Author Response

  • The quantity of audio samples exchanged was limited due to the privacy laws in the EU and the possible sensitivity of the data. Everything exchanged had to undergo multiple rounds of review to ensure no sensitive material was present.
  • "require special handling" corrected to "requires special handling"
  • "551words" corrected to "551 words"
  • "of words, i.e. the" corrected to "of words, i.e., the"
  • "facilities!" corrected to "facilities."
  • Removed redefinition of ASRU and replaced with "with and without ASRU support"
  • "determines ATCo’s workload" corrected to "determines an ATCo’s workload"
  • The "(CPDLC)" following the application name denotes the shorthand that was used to reference the project in Table VIII.
  • "future, ATCo and pilots" corrected to "future, ATCos and pilots"
  • "(when ATCo points" corrected to "(when an ATCo points"
  • "determine, if the arrival" corrected to "determine if the arrival"
  • Removed "we got lucky!"
  • "nor the CRER tell us" corrected to "nor the CRER tells us"
  • "we marked the words, which were" corrected to "we marked the words that were"
  • "by e.g. GDPR" corrected to "by, e.g., GDPR"
  • "to derive, whether it" corrected to "to derive whether it"

Reviewer 2 Report

This paper focuses on the language ontology of transatlantic ASRU in the ATM domain. The authors comprehensively presented the difference in language ontology between MITRE and DLR. Overall, this paper is interesting and insightful.  There are minor issues that should be further improved: 

1.  The contributions of this paper should be clearly pointed out in the introduction Section.

2. Fig. 6 is presented with poor quality. It should be improved in the revision.

 Minor editing of English language required

Author Response

  • We expanded on the last two sentences of subsection 1.1 Broad Context of the Study which originally stated: "This paper expands on that topic to discuss the impact of the ontology on future research and development collaboration. This paper also examines the word-level differences between United States and European ATC speech to provide quantitative understanding of the corpus data that feed the ASRU models, informing their potential cross use between regions." Our expansion elaborates to include findings and conclusions: "This paper expands on that topic to discuss the impact of the ontology on future research and development collaboration, describing several ways an ATC ontology is critical to facilitating collaboration between researchers and to appropriately evaluating ASRU applications in the ATM domain. This paper also examines the word-level differences between United States and European ATC speech to provide quantitative understanding of the corpus data that feed the ASRU models, informing their potential cross use between regions. The analysis shows that while the methods and tools for developing and measuring ASRU performance can be shared across regions (e.g., between US and Europe), the specific models built for the different regions would likely not work well across regions."
  • Do you mean the figure is poor quality and hard to read? If so, I have enlarged it and moved some paragraphs to improve page spacing. If you mean the content of the diagram is not described in detail, then this was a deliberate choice by the authors. Before the diagram, we state, "Figure 6 illustrates the effect of different errors during the automatic PIREP detection logic within the application and their effect on overall application performance." The diagram was then included to condense text describing the detailed cause-and-effect relationships between component error and system performance. Because of its inclusion, we felt comfortable summarizing the final results of different errors in a couple of sentences before the diagram. Are there parts of the diagram that are unclear or ambiguous?
